# Semi-supervised Vertex Hunting, with Applications in Network and Text Analysis

**Yicong Jiang**
Department of Statistics
Harvard University
yicong_jiang@g.harvard.edu

**Zheng Tracy Ke**
Department of Statistics
Harvard University
zke@fas.harvard.edu

## Abstract

Vertex hunting (VH) is the task of estimating a simplex from noisy data points and has many applications in areas such as network and text analysis. We introduce a new variant, semi-supervised vertex hunting (SSVH), in which partial information is available in the form of barycentric coordinates for some data points, known only up to an unknown transformation. To address this problem, we develop a method that leverages properties of orthogonal projection matrices, drawing on novel insights from linear algebra. We establish theoretical error bounds for our method and demonstrate that it achieves a faster convergence rate than existing unsupervised VH algorithms. Finally, we apply SSVH to two practical settings—semi-supervised network mixed membership estimation and semi-supervised topic modeling—resulting in efficient and scalable algorithms.

## 1 Introduction

Semi-supervised learning has been widely studied in the classification settings with discrete-valued labels. In contrast, continuous-valued labels (e.g., soft labels) also play crucial role in applications. One example is network mixed membership estimation [2]. Suppose a network contains $K$ communities, and each node has a mixed membership over $K$ communities. The mixed membership vectors take arbitrary values in the probability simplex of $\mathbb{R}^K$. Another example is topic modeling [7]. Suppose a text corpus contains $K$ topics. Each word has a $K$-dimensional topic loading vector representing its relevance to $K$ topics. These topic loading vectors can also be regarded as soft labels of the words.

We are interested in semi-supervised learning for soft labels, where a small fraction of them are known. In the previous network estimation example, we may know the mixed membership vectors for some nodes (for example, this happens in dynamic networks with both long-existing and newly emerging nodes, where the mixed membership of a long-existing node can be inferred from historical data). In topic modeling, we may have knowledge of the topic relevance of certain words, such as anchor words [5] or seed words [15] for some topics, which can be leveraged for model estimation.

Vertex hunting (VH), also called linear unmixing or archetypical analysis, is an important tool used in the above problems for unsupervised learning. Fix $K \geq 2$ and a simplex $\mathcal{S} \subset \mathbb{R}^d$ that has $K$ vertices $v_1, v_2, \ldots, v_K$. Let $w_1, w_2, \ldots, w_n \in \mathbb{R}^K$ be weight vectors (a weight vector is such that all entries are non-negative and sum to 1). Suppose we observe $x_1, x_2, \ldots, x_n \in \mathbb{R}^d$ satisfying

$$x_i = r_i + \epsilon_i, \qquad \text{where} \quad r_i = \sum_{k=1}^{K} w_i(k) v_k, \quad \text{and } \epsilon_i\text{'s are i.i.d. noise.} \tag{1}$$

Here, each $r_i$ is contained in the simplex $\mathcal{S}$, and $w_i$ is called the *barycentric coordinate* of $r_i$. The goal of VH is to estimate the vertices $v_1, \ldots, v_K$ from the noisy data cloud $\{x_i\}_{i=1}^n$. In both unsupervised mixed membership estimation and topic modeling, the spectral-projected data exhibit such simplex structure, so that VH is frequently used as a plug-in step in parameter estimation [1, 5, 21, 25].

39th Conference on Neural Information Processing Systems (NeurIPS 2025).

In this paper, we introduce the *semi-supervised vertex hunting (SSVH)* problem, as a new tool for semi-supervised learning in the above problems. Suppose for a subset $S \subset \{1, 2, \ldots, n\}$, we observe $\pi_i \in \mathbb{R}^K$ for each $i \in S$, where $\pi_i$ is related to the barycentric coordinate $w_i$ as follows:

$$w_i = (b \circ \pi_i)/\|b \circ \pi_i\|_1, \qquad \text{for an unknown positive vector } b \in \mathbb{R}^K. \qquad (2)$$

Here $\circ$ is the Hadamard (entrywise) product. In this expression, if we multiply $b$ by any positive scalar, the equality continues to hold. Therefore, we assume $\|b\| = 1$ without loss of generality. SSVH aims to estimate $v_1, \ldots, v_K$ from $\{x_i\}_{i=1}^n$ and the additional information $\{\pi_i\}_{i \in S}$.

Model (2) was discovered in the literature of unsupervised learning, such as [21] for mixed membership estimation and [25] for topic modeling. In these problems, the mean of data matrix admits a nonnegative factorization structure. Under such structures, for any low-dimensional linear projection of data (including the spectral projection), the projected points are contained in a simplicial cone subject to noise corruption. To enable downstream estimation procedures, we must first normalize this simplicial cone to a simplex (e.g., for spectral projections, the SCORE normalization [20] is a convenient choice), and Model (2) is a direct consequence of such normalizations [24]. See Section 4 for details, where we validate (2) for the two applications of interest. It is worth noting that Model (2) was often hidden in the proofs of the previous works but not explicitly presented there, due to that an unsupervised VH algorithm does not need any knowledge of how the barycentric coordinate $w_i$ is related to the true $\pi_i$. In contrast, for semi-supervised VH, the connection between $\pi_i$ and $w_i$ directly affects how we design the algorithm, so we must present Model (2) explicitly here.

Many algorithms have been developed for unsupervised VH, such as minimum volume transformation (MVT) [8], N-FINDR [32], and successive projection (SP) [4]. Based on MVT, [14] developed a delicate anchor-free topic model estimation approach, which can also be applied to the VH problem. Recently, [22] provided a refinement of SP to strengthen its robustness against noise; [34] estimated the vertices in unsupervised overlapping community detection via K-median clustering under certain asymptotic regime; [16] adopted a regularized negative matrix factorization (NMF) for vertex hunting (archetypal analysis); and [30] proposed a theoretical framework for interpretation and guidance on spectral methods for network membership estimation and algorithms for vertex hunting such as MVT. However, it is unclear how to modify these methods to incorporate the information within $\{\pi_i\}_{i \in S}$. The difficulty stems from that $b$ is unknown—hence the knowledge of $\pi_i$ does not directly imply the barycentric coordinate of $r_i$ inside the simplex. One may consider a joint-optimization approach, where we optimize over $b$ and $v_1, \ldots, v_K$ together using a loss function, but it is unclear how to design a loss function that both facilitates computation and comes with a theoretical guarantee.

We overcome the difficulty by proposing an optimization-free estimate of $b$: For any vector $\alpha \in \mathbb{R}^{|S|}$ satisfying mild conditions, we construct a $K \times K$ matrix $\widehat{M}(\alpha)$ and let $\widehat{b}$ be the eigenvector of this matrix associated with its smallest eigenvalue. This estimator is easy to implement and enjoys nice theoretical properties. Our method is inspired by non-trivial insight in linear algebra: the construction of $\widehat{M}(\alpha)$ carefully utilizes properties of orthogonal projection matrices.

Once $\widehat{b}$ is obtained, we can derive the barycentric coordinate $w_i$ for $i \in S$ at ease. It provides the locations of these $r_i$ inside the simplex, and we can utilize such information to enhance an existing unsupervised VH algorithm. In fact, given $\widehat{b}$, we can even use a simple regression to get $\hat{v}_1, \ldots, \hat{v}_K$. This gives our final SSVH algorithm.

We show that SSVH has several benefits compared to unsupervised VH: First, unsupervised VH needs strong identification conditions. For instance, SP requires that at least one $r_i$ is placed at each vertex, and MVT requires that the minimum-volume simplex containing $r_1, \ldots, r_n$ is unique. When such conditions are violated, unsupervised VH may fail. Second, the error rate of unsupervised VH does not decay with $n$ [11, 22], so it is unable to take advantage of having more data points. Third, the signal-to-noise ratio of unsupervised VH depends on the $(K-1)$th singular value of the vertex matrix [22]. When $K$ is large, this singular value can be small, indicating that the simplex is 'thin' in some direction and vulnerable to noise corruption; hence, unsupervised VH may have unsatisfactory performance for large $K$. SSVH can address these issues—as we will demonstrate, it requires weaker identification conditions, enjoys a fast-decaying error rate, and can handle large values of $K$.

We apply SSVH to semi-supervised mixed-membership estimation and semi-supervised topic modeling and develop new methods for these two problems. For the first problem, despite of many methods for semi-supervised community detection [26, 19, 33, 6, 17, 28, 35], they are hard to generalize to allow for mixed membership. For example, one strategy in such methods (e.g., see [19]) is to group

labeled nodes according to their true communities and compute the 'similarity' between an unlabeled node and each group. When there is mixed membership, it is unclear how to define groups and compute the similarity metrics. For the second problem, seeded topic modeling [15] and keyword-assisted topic modeling [10] can be regarded as semi-supervised learning methods. However, they add prior information as Dirichlet priors, not permitting specification of topic relevance for individual words. Additionally, these Bayesian approaches are computationally intensive for large corpora, while our SSVH-powered algorithms can run much faster in some settings.

In summary, we introduce the SSVH problem and make the following contributions:

- *Methodology*: We propose an SSVH algorithm, the core idea of which is an optimization-free approach to estimating $b$. Our method is inspired by delicate insight in linear algebra.
- *Theory*: We prove an explicit error bound for SSVH under sub-Gaussian noise. We show that the error bound decreases fast as the size of $S$ increases.
- *Application*: We apply SSVH to network mixed membership estimation and topic modeling, obtaining new semi-supervised learning algorithms for these two problems.

**Notations**: Write $N = |S|$. Let $W_S, \Pi_S \in \mathbb{R}^{N \times K}$, $R_S \in \mathbb{R}^{N \times d}$, and $X_S \in \mathbb{R}^{N \times d}$ be the matrices of stacking together the $w_i$, $\pi_i$, $r_i$, and $x_i$ for $i \in S$, respectively. Let $V \in \mathbb{R}^{K \times d}$ be the matrix whose $k$th row is equal to $v_k'$. With these notations, model (1) can be re-written as $R_S = W_S V$. By elementary linear algebra, an eigenvector of a matrix is defined up to any scalar multiplication. Throughout this paper, when we compute the eigenvector of a matrix, the default scalar multiplication is chosen such that the eigenvector has a unit $\ell^2$-norm and that the sum of its entries is positive.

## 2 Method for Semi-supervised Vertex Hunting

### 2.1 The oracle case

In the oracle case, we observe $r_i = \mathbb{E}[x_i]$ and aim to recover $V$ from $r_1, r_2, \ldots, r_n$ and $\pi_i$ for $i \in S$. The key is to find an approach for recovering $b$. Once $b$ is known, we can immediately use (2) to recover the barycentric coordinate $w_i$ for $i \in S$, and the problem becomes relatively easy.

We tackle the estimation of $b$ by an interesting discovery in linear algebra. We introduce an $N \times N$ matrix, which is the projection matrix to the orthogonal complement of the column space of $\Pi_S$:

$$H = H(S) = I_N - \Pi_S(\Pi_S'\Pi_S)^{-1}\Pi_S'. \tag{3}$$

For any vector $v$, let $\text{diag}(v)$ denote the diagonal matrix whose diagonal entries are from $v$. Given any $\alpha \in \mathbb{R}^N$, we construct a $K \times K$ matrix:

$$M(\alpha) = M(\alpha; S) = \Pi_S'\text{diag}(H\alpha)R_S R_S'\text{diag}(H\alpha)\Pi_S. \tag{4}$$

Our design of $M(\alpha)$ is based on a novel idea of leveraging properties of projection matrices. In the following theorem, we show that $M(\alpha)$ has a nice property:

**Theorem 2.1** (Main discovery). *For any $\alpha \in \mathbb{R}^n$, $M(\alpha)b = \mathbf{0}_K$. Therefore, $b$ is an eigenvector of $M(\alpha)$ associated with the zero egienvalue.*

*Proof of this theorem:* Let $J(\alpha) = R_S'\text{diag}(H\alpha)\Pi_S$. Then, $M(\alpha) = J(\alpha)'J(\alpha)$. It suffices to show $J(\alpha)b = \mathbf{0}_d$. First, model (1) implies $R_S = W_S V$. It follows that $J(\alpha)b = V'W_S' \cdot \text{diag}(H\alpha)\Pi_S b$. Second, model (2) implies $w_i = (\pi_i \circ b)/\|\pi_i \circ b\|_1$; in the matrix form, this can be expressed as $W_S = [\text{diag}(\Pi_S b)]^{-1}\Pi_S\text{diag}(b)$. We plug $W_S$ into $J(\alpha)b$ to obtain:

$$J(\alpha)b = V'\text{diag}(b)\Pi_S'[\text{diag}(\Pi_S b)]^{-1}\text{diag}(H\alpha)\Pi_S b$$

$$= V'\text{diag}(b)\Pi_S'\text{diag}(H\alpha)[\text{diag}(\Pi_S b)]^{-1}\Pi_S b \quad \text{(switching diagonal matrices)}$$

$$= V'\text{diag}(b)\Pi_S'\text{diag}(H\alpha)\mathbf{1}_N \quad \text{(because } \text{diag}(v)^{-1}v = \mathbf{1} \text{ for a vector } v)$$

$$= V'\text{diag}(b)\Pi_S'H\alpha. \quad \text{(because } \text{diag}(v)\mathbf{1} = v \text{ for a vector } v) \tag{5}$$

We recall that $H$ is the projection matrix to the orthogonal complement of $\Pi_S$. Hence, $\Pi_S'H$ is a zero matrix. It follows that the right hand side of (5) is a zero vector. □

Theorem 2.1 states that $b$ is an eigenvector associated with the zero eigenvalue of $M(\alpha)$. However, it does not imply that $b$ is the unique eigenvector associated with the zero eigenvalue. The uniqueness holds only if the null space of $M(\alpha)$ is a one-dimensional subspace. The next theorem provides a sufficient condition for the uniqueness to hold:

**Theorem 2.2** (Uniqueness). *Let $\bar{w}_* = \frac{1}{N} \sum_{i \in S} w_i$ and define a $K \times K$ matrix by*

$$\Sigma(\alpha) = \Sigma(\alpha, S) = \frac{1}{N} \sum_{i \in S} (H\alpha)_i (\pi_i' b) \cdot (w_i - \bar{w}_*)(w_i - \bar{w}_*)'. \tag{6}$$

*When* $\mathrm{rank}(\Sigma(\alpha)) = K - 1$*, the null space of $M(\alpha)$ is a one-dimensional subspace. Consequently, the eigenvector associated with the zero eigenvalue of $M(\alpha)$ is unique and must be equal to $b$.*

To see when the condition $\mathrm{rank}(\Sigma(\alpha)) = K - 1$ holds, we consider a simple case where $b = \mathbf{1}_K$ and $H\alpha = \mathbf{1}_n$ and give two examples. In the first example, there are $K + 1$ labeled points, with one point at each vertex and the last point located in the interior of the simplex (not on any vertex/edge/face). When the simplex is non-degenerate , $\Sigma(\alpha)$ has a rank $K - 1$. In the second example, the $\pi_i$'s of labeled nodes are i.i.d. sampled from a Dirichlet distribution, with the Dirichlet parameters being all positive constants. As $N \to \infty$, $\Sigma(\alpha)$ has a rank $K - 1$ with an overwhelming probability.

Inspired by Theorems 2.1-2.2, we obtain a method for recovering $b$ from the eigenvector of $M(\alpha)$. Once $b$ is known, by (2), $w_i$ is known for $i \in S$. We then have many options for recovering $V$. A simple method is the following regression approach. Recalling that $R_S = W_S V$, we recover $V$ from

$$V = (W_S' W_S)^{-1} W_S' R_S. \tag{7}$$

Another option is a penalized optimization approach. Let $L(V; r_1, \ldots, r_n)$ be a loss function that quantifies how well the simplex spanned by $V$ fits the data points. Such loss functions exist in many unsupervised VH algorithms (e.g., [8, 16]). We propose the following optimization:

$$\min_V \quad L(V; r_1, \ldots, r_n) + \lambda \sum_{i \in S} \|r_i - V w_i\|^2. \tag{8}$$

When $\lambda = \infty$, it reduces to the estimator in (7), which is the main version we will use. Meanwhile, one can always use a finite $\lambda$. Then, (8) and our method of estimating $b$ together offer an approach for extending any (optimization-based) unsupervised VH method to the semi-supervised setting.

**Remark 1**: Unsupervised VH requires identification conditions to uniquely determine the simplex from $r_1, r_2, \ldots, r_n$. For example, SP [4] requires that there is at least one $r_i$ locating at each vertex, and MVT [8] and anchor-free approaches such as [14] require that among all simplexes that contain $r_1, r_2, \ldots, r_n$, there is a unique one that minimizes the volume. When such conditions are violated, unsupervised VH may fail (see Figure 1 and more details in Appendix G). SSVH addresses this issue by leveraging the additional information in $\{\pi_i\}_{i \in S}$.

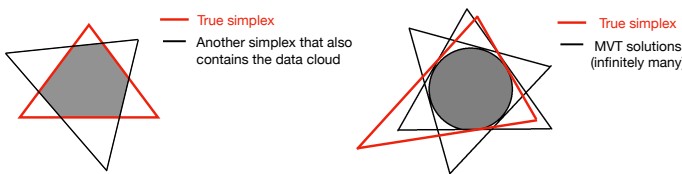

Figure 1: The identification issue for SP (left) and MVT (right). The grey area is the area covered by $r_1, \ldots, r_n$ (the noiseless point cloud). Left: There exists no $r_i$ on the vertices of the true simplex; consequently, there are multiple simplexes containing the point cloud, and the SP solution is not necessarily the true simplex. Right: the point cloud is a ball, and the MVT solution (the minimum-volume simplex that contains this ball) is not unique and does not include the true simplex.

## 2.2 The real case

In the real case, we observe $x_i$ instead of $r_i$. The oracle method can be extended: First, we construct $\widehat{M}(\alpha)$ by replacing $R_S$ in (4) by $X_S$. Due to noise corruption, this matrix often doesn't have a zero eigenvalue; but we can estimate $b$ by the eigenvector associated with the smallest eigenvalue. Second, we plug $\hat{b}$ into (7) and replace $R_S$ by $X_S$ there. This gives $\widehat{V}$. See Algorithm 1.

**Choice of $\alpha$**: In Algorithm 1, the only remaining question is how to choose $\alpha$. In the oracle case, as long as the matrix $\Sigma(\alpha)$ defined in (6) has a rank $K - 1$, any choice of $\alpha$ yields the precise recovery of $V$. In the real case, we still have a wide range of choice of $\alpha$, as long as the the signal-to-noise ratio (SNR) in $\widehat{M}(\alpha)$ is properly large. We offer two recommended approaches, both being closed-form.

Since re-scaling $\alpha$ doesn't alter $\hat{b}$, we set $\|\alpha\| = 1$. Note that $b$ and $\hat{b}$ are the eigenvector of $M(\alpha)$ and $\widehat{M}(\alpha)$ associated with the smallest eigenvalue, respectively. By sin-theta theorem [9], the eigen-gap,

---

**Algorithm 1:** Semi-supervised Vertex Hunting (SSVH)

---
1 Input: $K$, $X_S$, and $\Pi_S$.

  1. Compute $\alpha \in \mathbb{R}^N$ from $\Pi_S$ using the *closed-form* solution of either (9) or (10).

  2. Construct $\widehat{M}(\alpha) = \Pi'_S \text{diag}(H\alpha) X_S X'_S \text{diag}(H\alpha) \Pi_S$, where $H$ is as in (3). Let $\hat{b}$ be the eigenvector of $\widehat{M}(\alpha)$ corresponding to the smallest eigenvalue.

  3. Obtain $\hat{w}_i = (\hat{b} \circ \pi_i) / \|\hat{b} \circ \pi_i\|_1$, and let $\widehat{W}_S$ be the matrix of stacking together the $\hat{w}_i$ for $i \in S$. Compute $\widehat{V} = (\widehat{W}'_S \widehat{W}_S)^{-1} \widehat{W}'_S X_S$.

Output: $\widehat{V}$ (its rows are the estimated vertices).

---

$\lambda_{K-1}(M(\alpha))$, plays a key role in determining the signal strength. With $\|\alpha\| = 1$, we empirically observe that $\lambda_{K-1}(M(\alpha))$ often increases with $\|M(\alpha)\|$. It inspires us to maximize $\|M(\alpha)\|$ subject to $\|\alpha\| = 1$. This optimization still does not have an explicit solution. However, when the volume of the simplex is lower bounded (which means that the simplex is not 'super thin' in any direction), it holds approximately that $\|M(\alpha)\| \geq \omega \cdot \|\Pi'_S \text{diag}(H\alpha) \Pi_S\|_F^2$, where $\omega$ is a quantity that depends on model parameters but not $\alpha$ (this derivation is technical thus contained in the supplement). Therefore, we solve the following optimization:

$$\max \|\Pi'_S \text{diag}(H\alpha) \Pi_S\|_F^2, \qquad \text{s.t.} \|\alpha\| = 1 \qquad (9)$$

Without much effort, we can show that (9) has a **close-form** solution: Let $\Gamma_S = \Pi_S \Pi'_S$, and let $f(\Gamma_S)$ be the matrix of applying a univariate function $f(c) = c^2$ on $\Gamma_S$ entry-wisely. The solution $\alpha^*$ is the eigenvector of $H f(\Gamma_S) H$ corresponding to the largest eigenvalue.

Another option of choosing $\alpha$ uses dimension reduction. We divide $S$ into $K + 1$ non-overlapping clusters $\mathcal{C}_1, \ldots, \mathcal{C}_{K+1}$, by running k-means clustering on $\{\pi_i\}_{i \in S}$. Let $\pi_i^{\text{net}} = e_k$ (the $k$th standard basis vector of $\mathbb{R}^{K+1}$) for $i \in \mathcal{C}_k$. Let $\Pi_S^{\text{net}} \in \mathbb{R}^{N \times (K+1)}$ be the matrix of stacking together $\pi_i^{\text{net}}$ for all $i \in S$. Define $U^{\text{net}} = \Pi_S^{\text{net}}[(\Pi_S^{\text{net}})'\Pi_S^{\text{net}}]^{-1}(\Pi_S^{\text{net}})'$, the projection matrix into the column space of $\Pi_S^{\text{net}}$ (which is a subspace with dimension $\leq K + 1$). We solve the optimization:

$$\max \|M(U^{\text{net}}\alpha)\|, \qquad \text{s.t.} \|\alpha\| = 1. \qquad (10)$$

At first glance, this problem is not easier than maximizing $\|M(\alpha)\|$ directly. What makes a difference is that when $\Pi'_S \Pi_S^{\text{net}}$ has a full rank $K$, $HU^{\text{net}}$ projects all vectors into a 1-dimensional subspace. In this case, maximizing $\|M(U^{\text{net}}\alpha)\|$ is equivalent to maximizing $\|HU^{\text{net}}\alpha\|$, and the problem has a **closed-form** solution: $\alpha^*$ is the right eigenvector of $HU^{\text{net}}$ associated with the largest eigenvalue of $HU^{\text{net}}$. We remark that even when the column space of $HU^{\text{net}}$ is not 1-dimensional, we can still compute this closed-form $\alpha^*$, though it is no longer the solution of (10).

**Remark 2**: It is important to note that, although these choices of $\alpha$ are motivated by optimizations involving $M(\alpha)$, their closed-form solutions are only functions of $\Pi_S$. In other words, these $\alpha$ neither use noisy data nor depend on unknown model parameters. This is why we can legitimately impose regularity conditions on $\alpha$ in the theoretical analysis to be presented in Section 3.

## 3 Theoretical Results

In this section, we derive the error bound for our estimator in Algorithm 1. As we have mentioned, the scaling of $b$ and $\alpha$ doesn't matter. We assume $\|b\| = 1$ and $\|\alpha\| = 1$ without loss of generality.

**Assumption 3.1.** *There exist constants $c_1, c_2, c_3 > 0$, such that (a) $\|V\| \geq c_1$ and $\lambda_{K-1}(V) \geq c_1\|V\|$, (b) $\lambda_{min}(\Pi_S) \geq c_2 \cdot \lambda_{max}(\Pi_S)$, and (c) $\min_k b_k \geq c_3 \cdot \max_k b_k$.*

**Assumption 3.2** (Sub-Gaussian noise). *There exists $\sigma_n > 0$ (which may depend on $n$) such that $\mathbb{E}[\exp(t\epsilon_{ij})] \leq \exp(t^2\sigma_n^2/2), i \in S, j \in [K], t \in \mathbb{R}$.*

**Assumption 3.3.** *There exists a constant $c_4 > 0$ such that $\lambda_{K-1}(\Sigma(\alpha)) \geq c_4/(\sqrt{KN})$, where $\Sigma(\alpha)$ is the matrix defined in Theorem 2.2.*

Assumption 3.2 assumes that the noise within the data is sub-Gaussian, and the noise level $\sigma_n$ may grow as $n \to \infty$. Regarding Assumption 3.1, since the volume of the simplex is related to $\lambda_{K-1}(V)$, item (a) imposes a lower bound on the volume after re-scaling, which prevents the simplex to be too *thin* in some direction. Item (b) states that the labeled points are well spread-out in the simplex.

For instance, they can't concentrate in a small region, in which case the conditioning number of $\Pi_S$ is large. Item (c) implies that $\pi_i$ retains enough information of *each* coordinate of $w_i$ (e.g., when some $b_k$ is much smaller than the others, information of $w_i(k)$ is nearly lost in $\pi_i$). Assumption 3.3 is about the matrix $\Sigma(\alpha)$. In Theorem 2.2, we assume that its $(K-1)$th singular value is nonzero; and here we put a slightly stronger condition.

Our first result is a non-stochastic bound, which drops Assumption 3.2 and treats $\epsilon_i$ as non-stochastic:

**Lemma 3.1** (Non-stochastic bound)**.** *Suppose Models* (1)-(2) *and Assumptions 3.1, 3.3 hold. Define*
$$\text{err}_1 := \|\Pi_S' \text{diag}(H\alpha)(X_S - R_S)\|_F, \quad \text{err}_2 := \|W_S'(X_S - R_S)\|_F, \quad \text{err}_3 := \|X_S - R_S\|_{\max}.$$
*There exist constants $C_5, c_6 > 0$ that only depend on $c_1, \ldots c_4$, such that when $\text{err}_1 < c_6\sqrt{N/K}$,*
$$\|\widehat{V} - V\| \le \|V\| \cdot C_5 K[\text{err}_1(1 + \text{err}_3)\sqrt{N} + \text{err}_1^2/N + \text{err}_2/N]. \tag{11}$$

We use matrix spectral norm as the loss metric. The row-wise norm $\|\hat{v}_k - v_k\|$ is more frequently used in the literature [11, 22]. Our bound implies the row-wise bound because $\max_k \|\hat{v}_k - v_k\| \le \|\widehat{V} - V\|$.

The three error terms defined in Lemma 3.1 are functions of the 'noise' matrix $X_S - R_S$. Our next result provides high-probability bounds for them under sub-Gaussian noise.

**Lemma 3.2** (Noise terms)**.** *Suppose Models* (1)-(2) *and Assumptions 3.1, 3.2 hold. Suppose $N \ge K$. There exists a universal constant $C_{\text{err}}$ (which does not depend on any other constant in the assumptions) such that with probability at least $1 - 2/n = 1 - O(1/n)$,*
$$\text{err}_1 \le \sigma_n \sqrt{K + C_{\text{err}}^2 \sqrt{K} \log(n)}, \quad \text{err}_2 \le C_{\text{err}} \sigma_n \sqrt{N\sqrt{K}\log(n)}, \quad \text{err}_3 \le C_{\text{err}}\sqrt{\log(n)}. \tag{12}$$

Here the absolute constant $C_{\text{err}}$ depends on the absolute constants of Hanson–Wright inequality [31, Theorem 1.1]. [29] provides an evaluation of Hanson–Wright inequality's constants, based on which a rough estimate of $C_{\text{err}}$ is 2.8042. See Appendix D.4 for more details. Combining Lemma 3.1 and Lemma 3.2, we have the main theorem:

**Theorem 3.1.** *Suppose Models* (1) (2) *and Assumptions 3.1-3.3 hold. Suppose $N > (1/c_6^2) \cdot \sigma_n^2(K^2 + C_{\text{err}}^2 K^{1.5} \log(n))$, where $c_6$ is as in Lemma 3.1 and $C_{\text{err}}$ is as in Lemma 3.2. There exists a constant $\tilde{C}_5 > 0$ only depending on $c_1, \ldots c_4$, and $C_{\text{err}}$, such that with probability at least $1 - O(1/n)$,*
$$\|\widehat{V} - V\| \le \|V\| \cdot \tilde{C}_5 N^{-1/2} K^{1.25} \sigma_n \sqrt{\log(n)}(\sqrt{K} + \log(n))^{0.5}. \tag{13}$$

In Theorem 3.1, we require the number of labeled nodes $N$ to be at least of order $\sigma_n^2(K^2 + K^{1.5}\log(n))$. This condition is mild. First, when $K$ and $\sigma_n$ are constants, it only requires $O(\log(n))$ of labeled nodes, which is much smaller than $n$. Second, in many applications (see Section 4), $\sigma_n$ decreases fast with $n$. In such settings, even a finite number of labeled nodes are sufficient.

**Faster rate than unsupervised VH**: We focus on comparing with the results for successive projection (SP). [11] derived a non-stochastic bound in terms of $[1/\lambda_K(V)] \max_{1 \le i \le n} \|\epsilon_i\|$; and [22] derived an improved bound where $\lambda_K(V)$ is replaced by $\lambda_{K-1}(V)$. Their bounds are for the loss $\max_k \|\hat{v}_k - v_k\|$. Translating their results to our loss and using concentration inequalities for sub-Gaussian variables, we have (see [22] and (G.43) in Appendix G.1 for details):
$$\|\widehat{V} - V\| \le \|V\| \cdot CK\sigma_n\sqrt{\log(n)}. \tag{14}$$
Comparing (14) with (13), we observe that the error rate of SSVH has an additional factor of $N^{-1/2}$ (ignoring $K^{0.25}$ and logarithmic terms), which converges faster than the error of unsupervised VH.

**Tightness of the error bound**: We consider an ideal situation where $b$ is given. It follows by (2) that $W_S$ is immediately known. Define a regression estimator (under Gaussian noise, this is the MLE):
$$\widehat{V}^* = (W_S'W_S)^{-1}W_S'X_S. \tag{15}$$
If the error rate in (13) matches with the rate of this estimator, then we believe that the rate is already sharp. The lemmas below shows that the error rate in (13) matches the ideal estimator's error rate only up to a $(\sqrt{K} + \log(n))^{0.5}$ factor, suggesting the tightness of our rate.

**Lemma 3.3** (Ideal estimator)**.** *Under the assumptions of Theorem 3.1, the ideal estimator satisfies that with probability at least $1 - O(1/n)$, $\|\widehat{V}^* - V\| \le \|V\| \cdot \tilde{C}_5 N^{-1/2} K^{1.25} \sigma_n \sqrt{\log(n)}$.*

**Remark 3** *(Noisy labels)*: Our theory can be extended to the case with incorrect or noisy labels. Let $\Pi_S$ and $\widehat{\Pi}_S$ be the true and noisy label matrices, respectively. In this case, there will be an extra term in the error bound, $N^{-1/2}\|\widehat{\Pi}_S - \Pi_S\|$. Notably, this term will not explode as $N$ increases.

# 4 Applications to Network Analysis and Text Analysis

## 4.1 Semi-supervised mixed membership estimation

Mixed membership estimation (MME) [2] is a problem of interest in network analysis. Let $A \in \mathbb{R}^{n \times n}$ be the adjacency matrix of an undirected network with $n$ nodes. The network has $K$ communities. Each node $i$ has a mixed membership vector $\pi_i \in \mathbb{R}^K$, where $\pi_i(k)$ is this node's fractional weight on community $k$. The degree-corrected mixed membership (DCMM) model [21] has been introduced:

**Definition 4.1** (DCMM model). *The upper triangle of $A$ contains independent Bernoulli variables, with $\mathbb{P}(A_{ij} = 1) = \theta_i \theta_j \cdot \pi_i' P \pi_j$, where $\pi_i \in \mathbb{R}^K$ is the mixed membership vector, $\theta_i > 0$ is the degree parameter, and $P \in \mathbb{R}^{K \times K}$ is a symmetric nonnegative matrix that has unit diagonal entries.*

**Definition 4.2** (Semi-supervised MME). *Suppose the network follows a DCMM model. Let $S$ be a subset of $\{1, 2, \ldots, n\}$. Given $A$ and $\pi_i$ for $i \in S$, we aim to estimate all $\pi_i$ for $i \notin S$.*

We propose the following algorithm. It first uses a full-rank matrix $U$ to project each column of $A$ to a $K$-dimensional vector $\tilde{x}_i = U'Ae_i$, and then uses a vector $\eta$ to normalize it: $x_i = \tilde{x}_i / (\eta' \tilde{x}_i)$. When $U$ consists of the first $K$ eigenvectors of $A$ and when $\eta = e_1$, the vectors $x_i$ coincide with the spectral projections in [21]. However, our approach permits a general choice of $U$ and $\eta$. For example, we may apply a community detection algorithm on $A$ to get an estimated community membership matrix $\widehat{\Pi}^{\mathrm{cd}} \in \mathbb{R}^{n \times K}$, where each row takes values in $e_1, \ldots, e_K$. Then, we can take $U = \widehat{\Pi}^{\mathrm{cd}}$ and $\eta = \mathbf{1}_K$. Under mild regularity conditions on $(U, \eta)$, we can show that up to noise corruption, $x_1, \ldots, x_n$ are contained in a simplex with $K$ vertices (see Theorem 4.1 below). Therefore, we apply SSVH to $x_i$'s.

---

**Algorithm 2:** Semi-supervised Mixed Membership Estimation

---

1 Input: $K$, $A$, and $\Pi_S$. Algorithm parameters: a matrix $U \in \mathbb{R}^{n \times K}$ and a vector $\eta \in \mathbb{R}^K$.

    1. Compute $x_i = U'Ae_i / (\eta'U'Ae_i)$ for $1 \leq i \leq n$. Let $X = [x_1, \ldots, x_n]' \in \mathbb{R}^{n \times K}$ and let $X_S \in \mathbb{R}^{N \times K}$ be the matrix of stacking the $x_i$ for $i \in S$.

    2. (SSVH). Apply Algorithm 1 to $(K, X_S, \Pi_S)$ to obtain $\widehat{V}$ and the intermediate quantity $\hat{b}$.

    3. Let $\widehat{B} = \mathrm{diag}(\hat{b})\widehat{V}$. For each $i \notin S$, compute $\widetilde{\pi}_i = e_i'AU\widehat{B}'(\widehat{B}\widehat{B}')^{-1}$. Let $\hat{\pi}_i$ be the vector by setting the negative entries in $\widetilde{\pi}_k$ to zero and re-normalizing to have a unit $\ell^1$-norm.

Output: $\hat{\pi}_i$ for $i \notin S$.

---

We justify this algorithm using the following theorem. Under the DCMM model, $\mathbb{E}[A] = \Omega - \mathrm{diag}(\Omega)$, where $\Omega$ is a matrix with $\Omega_{ij} = \theta_i \theta_j \cdot \pi_i' P \pi_j$. We call $\Omega$ the 'signal' matrix.

**Theorem 4.1** (Validity of the algorithm). *Suppose that the rank of $\Omega$ is $K$, and $(U, \eta)$ satisfies that $\mathrm{rank}(U) = K$ and $\eta'U'\Omega e_i > 0$ for all $i$. The following statements are true:*

    *(a) Let $r_i = U'\Omega e_i / (\eta'U'\Omega e_i)$, for $1 \leq i \leq n$. There exist $v_1, \ldots, v_K \in \mathbb{R}^K$ and a positive vector $b \in \mathbb{R}^K$ such that each $r_i = \sum_{k=1}^{K} w_i(k)v_k$, where $w_i = (b \circ \pi_i)/\|b \circ \pi_i\|_1$.*

    *(b) If we plug $(K, \Omega, \Pi_S)$ into Algorithm 2, then $\hat{\pi}_i = \pi_i$ for all $i \notin S$.*

The first statement in Theorem 4.1 justifies that Models (1)-(2) hold for $x_i$'s, so that our SSVH framework is applicable. (Here, the noise is $\epsilon_i = x_i - r_i \approx U'(A - \Omega)e_i / (\eta'U'\Omega e_i)$, which can be shown sub-Gaussian, as $(A - \Omega)e_i$ contains independent centered Bernoulli variables.) The second statement says that the proposed algorithm can exactly recover unknown $\pi_i$ in the noiseless case.

Compared with the unsupervised MME methods such as MSCORE [21] or [34], a major benefit of the above algorithm is that it doesn't require existence of pure nodes for each community. Additionally, Algorithm 2 is also a new unsupervised MME algorithm, if we replace SSVH by unsupervised VH in Step 2. The resulting algorithm is different from MSCORE [21] in the unsupervised setting.

## 4.2 Semi-supervised topic modeling

Topic modeling (TM) [7] is a widely used tool for text analysis. Suppose we have a corpus with $n$ documents written on a vocabulary of $p$ words. Let $Y = [Y_1, \ldots, Y_n] \in \mathbb{R}^{p \times n}$ be such that $Y_i(j)$ is the count of word $j$ in document $i$. Topic modeling aims to estimate $K$ topics from the corpus, where each topic is a distribution over vocabulary words, represented by a probability mass function (PMF) $A_k \in \mathbb{R}^p$. The *probabilistic Latent Semantic Indexing (pLSI)* model [12] is a common topic model:

**Definition 4.3** (pLSI model). *The word count vectors $Y_1, \ldots, Y_n$ are independent of each other, with $Y_i \sim \text{Multinomial}(N_i, \Omega_i)$, where $N_i$ is the length of document $i$, and $\Omega_i \in \mathbb{R}^p$ is a PMF satisfying that $\Omega_i = \sum_{k=1}^{K} \gamma_i(k) A_k$, with $\gamma_i(k)$ being the fractional weight that document $i$ puts on topic $k$.*

The unsupervised topic modeling aims to estimate $A = [A_1, \ldots, A_K]$ from the data matrix $Y$, and is addressed various algorithms via optimization or spectral methods [14, 25]. We now formulate the semi-supervised problem by discussing what 'additional information' means in practice. For each $1 \le j \le p$, let $a_j = (A_1(j), \ldots, A_K(j))' \in \mathbb{R}^K$. This vector contains word $j$'s frequencies in different topics. The *topic loading vector* for word $j$ is defined as $a_j^* = a_j / \|a_j\|_1$. Since the effect of word frequency heterogeneity has been removed in $a_j^*$, this vector purely captures word $j$'s topic relevance. For example, when $j$ is an anchor word [5, 25] of topic $k$, $a_j^*$ is equal to $e_k$. When $a_j^* = (0.5, 0.5, 0, \ldots)'$, it means that word $j$ is only related to the first two topics. Motivated by these observations, we formulate the semi-supervised problem as follows:

**Definition 4.4** (Semi-supervised TM). *Suppose the word count matrix $Y$ follows the pLSI model. Let $S$ be a subset of $\{1, 2, \ldots, p\}$. Given $Y$ and $a_j^*$ for $j \in S$, we aim to estimate $A$.*

There is literature on incorporating prior knowledge or human input into topic modeling [3, 15, 27, 10]. A common assumption [15, 10] is that a set of keywords is given for each topic, and the topic vector is modeled as $A_k = \pi_k \widetilde{A}_k + (1 - \pi_k) A_k^*$, where $\widetilde{A}_k$ and $A_k^*$ are the keyword-assisted and standard topics, respectively, and the support of $\widetilde{A}_k$ is restricted on the keyword set. All parameters $(\pi_k, \widetilde{A}_k, A_k^*)$ are estimated in a Bayesian framework using beta and Dirichlet priors. In comparison, our problem in Definition 4.4 permits a more flexible way of specifying the topic relevance of each keyword. If in $a_j^*$ we put an equal weight on each topic word $j$ serves as a keyword, then our framework is similar to those in the literature. But we can also put unequal weights, to incorporate more human knowledge on keywords. Another benefit of our framework is that we can leverage SSVH to get a fast algorithm, without relying on the sampling procedure in the Bayesian framework.

We now propose our algorithm for semi-supervised TM. The strategy is similar to that in Section 4.1. We first turn the counts to frequencies: $D = [d_1, d_2, \ldots, d_n]$, with $d_i = N_i^{-1} Y_i$. We use a matrix $U \in \mathbb{R}^{p \times K}$ to project each row of $D$ and a vector $\eta \in \mathbb{R}^K$ for normalization, and apply SSVH. After $\widehat{V}$ is obtained, how to estimate $A$ requires some derivation. We relegate details to the supplement and only present the algorithm ($A_S^*$ is the $N \times K$ matrix of stacking together those $a_j^*$ for $j \in S$):

---

**Algorithm 3:** Semi-supervised Topic Modeling

1 Input: $K$, $D$, and $A_S^*$. Algorithm parameters: a matrix $U \in \mathbb{R}^{p \times K}$ and a vector $\eta \in \mathbb{R}^K$.

    1. Compute $x_j = U'D'e_j / (\eta'U'D'e_j)$ for $1 \le j \le p$. Let $X = [x_1, \ldots, x_p]' \in \mathbb{R}^{p \times K}$ and let $X_S \in \mathbb{R}^{N \times K}$ be the matrix of stacking the $x_i$ for $i \in S$.

    2. (SSVH). Apply Algorithm 1 to $(K, X_S, A_S^*)$ to obtain $\widehat{V}$ and the intermediate quantity $\hat{b}$.

    3. Let $\widehat{B} = \text{diag}(\hat{b})\widehat{V}$. Estimate $A$ by $\widehat{A} = DU\widehat{B}'(\widehat{B}\widehat{B}')^{-1}$

Output: $\widehat{A}$ (its columns are the estimated topic vectors).

---

**Theorem 4.2** (Validity of the algorithm). *In the pLSI model, write $\Gamma = [\gamma_1, \ldots, \gamma_n]$ and $D_0 = A\Gamma$. Suppose the rank of $A\Gamma$ is $K$, and $(U, \eta)$ is such that $\text{rank}(U) = K$ and $\eta'U'D_0'e_j > 0$ for $1 \le j \le p$. If we plug $(K, D_0, A_S^*)$ into Algorithm 3, then $\widehat{A} = A$.*

## 5 Empirical Study

We evaluate the performance of our method, compare it with the unsupervised SP algorithm, and apply our method to the problems in Section 4. By default, we choose $\alpha$ in our algorithm using the first approach there. We use the loss $\min_P \|P\widehat{V} - V\|_F / K$, where minimum is over row permutations.

**Simulations:** We fix $n = 1000$ and generate $b$ by first sampling its $K$ entries independently from $\text{Uniform}(0.9, 1.1)$ and then normalizing it so that $\|b\| = 1$. The diagonal elements of $V$ are 1, and the off-diagonal entries are independently generated from $\text{Uniform}(0, 1/K)$ (we make the off-diagonal elements of $V$ less than $1/K$ is to guarantee $\lambda_{K-1}(V) = O(1)$ when $K$ is large). We consider a total of 5 experiments by varying the label ratio $|S|/n$, noise level $\sigma$, and dimension $K$, and 2 additional experiments comparing with unsupervised VH and studying the runtime.

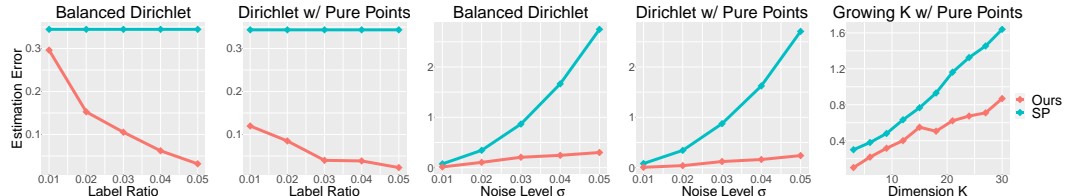

Figure 2: The influence of label ratio $N/n$, noise level $\sigma$, and dimension $K$ on error $\|\hat{V} - V\|_{\mathcal{F}}^2/K$. "Balanced Dirichlet" and "Dirichlet w/ Pure Points" correspond to setting (1) and (2) respectively.

**Experiments 1-2, influence of data points distribution**. We set $(K, \sigma) = (3, 0.2)$ and consider two different settings for $W$: (1) all the rows of $W$ are generated independently from Dirichlet$(1/3, 1/3, 1/3)$; (2) with $k = 1, 2, ..., K$, generate 2 rows of $W_S$ independently from the transformed Dirichlet distribution $0.1$Dirichlet$(1/3, 1/3, 1/3) + 0.9e_k$ so that they are almost pure, set 30 rows of $W_{S^C}$ to be $e_k$ so that they are pure, and finally generate all the other rows of $W$ independently from Dirichlet$(1/3, 1/3, 1/3)$. Setting (1) is more natural, but it does not guarantee the existence of pure points (i.e., $r_i$ is equal to one vertex). In setting (2), we purposely add pure points. Once $W$ is generated, we generate $x_i$'s following Models (1)-(2). In these two experiments, we vary the label ratio $|S|/n$ from 1% to 5%. The results based on 100 repetitions are given in Figure 2. In both experiments, with only very low label ratio, our algorithm can greatly outperform the unsupervised VH algorithm, successive projection (SP).

**Experiments 3-5, influence of noise level and data dimension**. In experiment 3 and 4, we fix $(K, |S|/n) = (3, 0.03)$ and vary the noise level $\sigma$. We still consider the two ways of generating $W$ as in Experiments 1-2. As seen in Figure 2, as the noise level increases, the error of our method grows much slower than that of SP. In experiment 5, we study the influence of $K$. Fix $\sigma = 0.2$. As $K$ increases, the number of vertices grows and it is reasonable to require more labeled points. We set $|S| = 4K$ (e.g., when $K = 3$, the label ratio $|S|/n = 0.012$). For large $K$, it is likely that some vertices are far from all the observed points, rendering difficulty in identifying them. To prevent this, we generate $W$ in the following way. With $k = 1, 2, ..., K$, we generate 2 rows of $W_S$ independently from the transformed Dirichlet distribution $0.1$Dirichlet$(\mathbf{1}_K) + 0.9e_k$, generate 2 rows of $W_S$ independently from $0.1$Dirichlet$(\mathbf{1}_K) + 0.9\mathbf{1}_K/K$, set 10 rows of $W_{S^C}$ to be $e_k$, and finally generate all the other rows of $W$ independently from Dirichlet$(\mathbf{1}_K)$. As seen in Figure 2, despite the high dimensionality, our method outperforms SP with only very few labeled points.

**Experiments 6, comparison with MVT and AA**. In this experiment, we compared our method with two NMF-based unsupervised methods: the minimum volume transformation (MVT, [8]) and the archetypal analysis (AA) algorithm [16]. Different from SP, both approaches are anchor-free: MVT can leverage the data points on the boundary to locate the minimum-volume simplex, and AA focus on the convex hull of the data cloud, minimizing the well-constructed distance between it and the estimated vertices. However, the two methods are restricted to utilizing only the convex hull information and rely on the assumption that the data points are scattered widely enough for the convex hull to cover most of the underlying simplex. Our method, conversely, can appropriately extract information from both the inner and boundary points. In this experiment, we fix $(K, |S|/n) = (3, 0.03)$ and vary the noise level $\sigma$. Because both MVT and AA are anchor free, we adopt the same generation process of $W$ as in Experiment 1 so that there may not exist no pure nodes. The results based on 100 repetitions are exhibited in Table 1. It illustrates that without leveraging the pure nodes, our method can outperform both of MVT and AA under various noise level.

Table 1: The median estimation error for SP, MVT, AA, and our method over 100 repetitions.

| $\sigma$ | 0.2 | 0.4 | 0.6 | 0.8 | 1 |
|---|---|---|---|---|---|
| Ours | **0.053** | **0.064** | **0.231** | **0.297** | **0.416** |
| SP | 0.319 | 1.712 | 4.438 | 8.404 | 13.37 |
| MVT | 0.191 | 0.38 | 0.39 | 0.427 | 0.522 |
| AA | 1.863 | 4.095 | 7.8 | 12.521 | 18.925 |

**Experiments 7, runtime analysis**. In this experiment, we study the runtime of SP, MVT, and our method as the dimension $K$ grows. We adopt the same setting as in experiment 5. The results aggregated from 100 repetitions are shown in Table 2. From the table, one can see that our method is very computationally efficient compared to unsupervised approaches such as SP or MVT. Additionally,

Table 2: The mean runtime (in milliseconds) of SP, MVT, and our method over 100 repetitions. MVT is very time-consuming for large $K$, so the results are omitted.

| K | 3 | 6 | 9 | 12 | 15 | 18 | 21 | 24 | 27 | 30 |
|---|---|---|---|---|---|---|---|---|---|---|
| Ours | 3.22 | 3.22 | 3.45 | 3.81 | 3.71 | 3.98 | 4.1 | 5.44 | 4.92 | 4.52 |
| SP | 5.11 | 9.62 | 10.44 | 17.67 | 24.44 | 32.58 | 35.93 | 44.67 | 46.11 | 53.55 |
| MVT | 8489 | >1e4 | >1e4 | >1e4 | >1e4 | >1e4 | >1e4 | >1e4 | >1e4 | >1e4 |

as $K$ scales up, the time complexity of unsupervised methods grows rapid, while our method maintains a low computation cost.

**Network mixed-membership estimation.** We use a co-authorship network for statisticians [18], with 2831 nodes and 71432 edges. The node degrees range from 2 to 853, showing with severe hetero-geneity. Since there is no ground-truth membership, we conduct a *semi-synthetic* experiment where we first apply Mixed-SCORE [21] to cablirate the parameters $K, P, \pi, \theta$ for DCMM and then generate synthetic networks with the estimated parameters. We set the label ratio $N/n = 0.05$ and compare our algorithm with two unsupervised algorithm, SP and a de-noised variant of SP called SVS [21] (it has a tuning parameter $L$, which is set to $L = 10 \times K$). The median loss on estimating $V$ over 100 repetitions is given in Table 3, and the estimated simplexes by different methods in one repetition are shown in Figure 3 (the points $x_i$'s are obtained using the spectral projections in [21]). Additionally, we also evaluated the excess error of plugging in different VH algorithms into Algorithm 2 for estimating the mixed membership vectors compared with the ideal case where the vertices are known (the loss is the Frobenius error $\|\hat{\Pi} - \Pi\|_F / \sqrt{nK}$). In Table 3, we report the Excess Error: the loss by plugging in our SSVH estimate minus the loss by plugging in the true simplex.

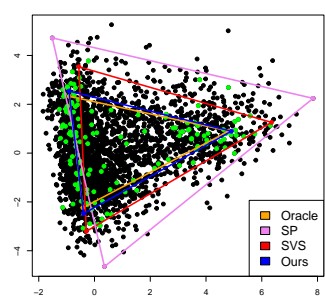

Figure 3: Comparison of the true simplex (orange), SSVH estimate (blue), SP estimate (pink), and SVS estimate (red). The green points are the labeled ones.

Table 3: The median estimation error of the vertices for SP, SVS, and our method over 100 repetitions. The numbers in parentheses denote median absolute deviation [13], a robust statistics for variability.

| | Error in $V$ | | | Excess Error of $\Pi$ or $A$ | | |
|---|---|---|---|---|---|---|
| | SP | SVS | Ours | SP | SVS | Ours |
| Network | 2.37 (0.26) | 1.05 (0.14) | **0.16 (0.04)** | 0.15 (0.024) | 0.08 (0.0083) | **0.02 (0.0035)** |
| Text Analysis | 1.77 (0.12) | 1.64 (0.11) | **1.22 (0.46)** | 0.052 (0.014) | 0.033 (0.0025) | **0.031 (0.0047)** |

**Topic modeling.** We use the academic abstracts in MADStat [23]. The processed word count provided by authors use a vocabulary of 2106 words. We further restrict to those abstracts whose total count on these 2106 words is at least 100. This results in 4129 documents. We fix $K = 11$ (following [23]) and apply the algorithm in [25] to calibrate model parameters $(A, W)$. We then generate synthetic corpus matrices. We set the label ratio $N/n = 0.05$ and compare our algorithm with two unsupervised algorithms, SP and SVS. Since $K$ is large, we set $L = K + 5$ in SVS (the time complexity of the algorithm scales with $\binom{L}{K}$). We perform 100 repetition of the semi-synthetic experiments, and compares the average estimation error of the vertices $\|\hat{V} - V\|_{\mathcal{F}} / K$ (up to permutation) for the three different methods. The results are displayed in Table 3. It can be seen that with only a small proportion of the label information, our method can greatly outperform the unsupervised methods. We also computed the Excess Error of plugging these algorithms into Algorithm 3 for estimating $A$ (the loss is Frobenius error $\|\hat{A} - A\|_F / K$), and the results are in Table 3.

Besides SP ans SVS, we also compare our method with a novel unsupervised topic model estimation approach [14]. We use the same synthetic dateset and setting as the previous empirical study except setting the number of labeled nodes $N = K + 1$. The prior information within this scenario is extremely weak, barely equipping us with any knowledge above the unsupervised setting. Remarkably, over 100 repetitions, our method's median estimation error in V is 0.023 , while the corresponding error of [14] is 0.030, which is 30.4% higher than us. This illustrates the efficiency of our algorithm even with exceedingly low signal. We also implemented SeededLDA [15], but the resulting error in $A$ is very large (about 40 times larger than our method). The main reason is that seeded LDA assumes a different model and is less valid in our scenario.

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
