# OpenReview forum: "Semi-supervised Vertex Hunting, with Applications in Network and Text Analysis"
_NeurIPS.cc/2025/Conference — NeurIPS 2025 poster_

### Official Review · Reviewer_AVSi · 2025-06-10

**Clarity:** 3
**Significance:** 2
**Originality:** 3
**Rating:** 4
**Confidence:** 3

**Summary:**

This paper introduces a novel problem formulation called Semi-supervised Vertex Hunting (SSVH), a variant of the classical vertex hunting (VH) problem, which seeks to estimate the vertices of a simplex from noisy data points. In SSVH, the semi-supervised setting assumes partial side-information about some points' barycentric coordinates, though only up to an unknown element-wise scaling (Hadamard product with an unknown positive vector b).

The authors propose an optimization-free method for estimating b, based on a clever construction of an eigenvalue problem leveraging projection matrix properties. Once b is estimated, existing VH methods can be enhanced by incorporating the supervised points.

Theoretical contributions include an explicit error bound for SSVH under sub-Gaussian noise, showing faster convergence compared to existing unsupervised VH approaches. Practical applications to semi-supervised mixed-membership estimation in networks and semi-supervised topic modeling are demonstrated, with empirical results on synthetic and semi-synthetic datasets showing significant improvements over baselines.

**Questions:**

The method is claimed to handle large K better than SP and MVT, but no wall-clock time comparisons or runtime analysis is provided. Could the authors report the computational cost (both theoretically and empirically) as K increases?

Section 2.2 proposes two methods to select α, but the empirical section always uses the first approach. Could the authors provide a comparison between the two choices to validate whether one is clearly preferable or whether both are robust?

In applications, the quality of the semi-supervised labels may vary — especially if derived heuristically. Could the authors study the impact of noisy or imperfect side information on SSVH's performance?

The paper focuses on networks and text analysis. Are there other promising domains where SSVH could be applied? A discussion of potential broader applicability would be appreciated.

The experimental comparison mainly uses SP and SVS. Could the authors also compare against modern NMF-based VH variants to further demonstrate superiority?

**Ethical Concerns:**

["NO or VERY MINOR ethics concerns only"]

**Limitations:**

The authors explicitly discuss limitations in Appendix. Moreover, the theoretical analysis clarifies conditions under which the method works well (Assumptions 3.1–3.3), and the empirical results demonstrate that the method is not brittle. Some points that could be highlighted even more explicitly:

The method depends on the spread and coverage of the labeled set S. If S is highly unbalanced or missing key regions of the simplex, performance may degrade.

Assumes the model of barycentric coordinate transformation (Eq. 2) holds; when this model is violated, behavior is not analyzed.

**Paper Formatting Concerns:**

The paper formatting generally follows NeurIPS guidelines well.

**Quality:**

2

**Strengths And Weaknesses:**

Strengths:

The paper is technically strong, with a novel and theoretically sound contribution. The idea of using an optimization-free estimation of b via properties of projection matrices is elegant and well-justified. Proofs are complete and provided in the supplementary material.

The paper is very clearly written. The introduction does an excellent job motivating the problem and situating it within both VH literature and applications in network and text analysis. The key technical insights are well-explained.

The proposed SSVH method addresses key limitations of existing VH methods—sensitivity to identification conditions, inability to leverage semi-supervised labels, and poor scaling in K. The paper shows strong potential impact for network community analysis and topic modeling — two high-visibility applications.

The work is original in both its problem formulation (semi-supervised vertex hunting with transformed barycentric coordinates) and its methodological solution (optimization-free eigen-based estimation). To my knowledge, no prior work provides this perspective on integrating side information into VH.

Weaknesses:

The empirical section, while providing clear gains, could be strengthened by including more real-world datasets. Both network and text experiments rely on semi-synthetic settings where the ground truth is known through re-simulation; it would be valuable to show performance on unmodified real datasets where only partial labels are available.

The paper mentions that the SSVH method can be combined with optimization-based VH, but this idea is left as future work. A small experiment on this combination would strengthen the practical value.

The dependence of the error bound on the choice of α is somewhat under-explored empirically. While two closed-form solutions are proposed, it would help to see how sensitive performance is to these choices.

---

> ### Author Rebuttal · Authors · 2025-07-31
>
> Thank you for your helpful comments. We are glad that you think our paper has a "novel and theoretically sound contribution" and our optimization-free estimation is "elegant and well-justified." Please find below our point-to-point response to your questions.
>
> **1. Runtime analysis.**
>
> Following your suggestion, we documented the runtime of our method as $K$ increases: Fix $n=1000$. For a grid of $K$, we ran our algorithm for 20 repetitions and divided the total runtime by 20. This gives the average runtime per iteration. For comparison, we have also included the runtime of successive projection (SP) for all $K$ and the runtime of minimum volume transformation (MVT, a classical unsupervised method) for $K=3$. All three methods are implemented on R.
>
> | K  | 3 | 6 | 9 | 12 | 15 | 18 | 21 | 24 | 27 | 30 |
> | -- | -- | -- | -- | -- | -- | -- | -- | -- | -- | -- |
> | Ours (millisecond) | 3.22 | 3.22 | 3.45 |  3.81 | 3.71 | 3.98 | 4.10 | 5.44 |  4.92 | 4.52 |
> | SP (millisecond) | 5.11 | 9.62 | 10.44 |  17.67 | 24.44 | 32.58 | 35.93 | 44.67 |  46.11 | 53.55 |
> | MVT (millisecond) | 8489 |
>
> Our method is super fast, and the runtime is insensitive to $K$. SP is also very fast, but its runtime increases considerably as $K$ increases. MVT requires solving a non-convex optimization via iterations. We used the CVRX package, via a convex relaxation of the objective, to implement MVT. It is much slower than our method and SP.
>
> The complexity of our method is as follows: $O(N^3)$ for finding $\alpha$; $O(NK^2+K^3)$ for estimating $b$; and $O(nKd + K^3)$ for obtaining $\widehat{V}$ from $b$, where $N$ is the number of labeled points and $d$ is the dimensionality of all points. When $K\leq N\ll n$, the total complexity is $O(NK^2+nKd)$.
>
>
> **2. Selection of $\alpha$.**
>
> *"Section 2.2 proposes two methods to select α, but the empirical section always uses the first approach. Could the authors provide a comparison between the two choices to validate whether one is clearly preferable or whether both are robust?"*
>
> This is a great point. We first answer your questions using a simulation experiment. We fix $n=1000$, $K=3$, $d=2$, $\sigma=0.2$ (noise standard deviation), and set the labeling ratio to 5%. We implement our method with three choices of $\alpha$: Approach 1 and Approach 2 in the paper, and a randomly perturbed version of Approach 1by adding $0.1\cdot \mathrm{Unif}(0, 1/\sqrt{N})$ noise to each entry of $\alpha$ (since the original $\alpha$ is a unit-norm vector with $N$ entries, such a perturbation is significant). The media errors over 20 repetitions are as follows:
>
> | $\alpha$-choice | Approach 1 | Approach 2 | Approach 1(perturbed) |
> | -- | -- | -- | -- |
> | Error | 0.0260 | 0.0229 | 0.0265 |
>
> In this example, there is no big difference between the two approaches. Additionally, the results are robust to perturbation of $\alpha$.
>
> Second, we explain why we presented two approaches.
> Approach 2 was inspired by our theoretical insight for the special case of $N=K+1$. In this case, we can explicitly find an optimal $\alpha$.  Such an $\alpha$ is not defined for $N>K+1$, but we hope to mimic the case of $N=K+1$ by grouping labeled points into $(K+1)$ groups. Although this idea was clear, we found that its performance relied on that labeled points were well-clustered. To deal with the case that labeled points are scattered, we designed Approach 1, which is more ad-hoc but works for broader cases.
>
> As for practice, our recommendation is to first run k-means on the labeled points and check whether they are well-clustered (e.g., by checking the Rayleigh quotient) . If yes, apply Approach 2; otherwise, always use Approach 1.
>
> **3. The impact of noisy or imperfect side information.**
>
> Yes, our theory can incorporate this. Let $\Pi_S$ and $\widehat{\Pi}_S$ be the true and noisy (or imperfect) label matrices for those labeled points. There will be an extra term in the error bound (the norm below is the spectral norm):
> $$N^{-1/2}\| \widehat{\Pi}_S-\Pi_S \|.$$
>
> Notably, this term will not explode as $N$ increases, so our method still enjoys a good error rate.
>
> **4. Comparison with modern NMF-based VH variants.**
>
> We added two competitors:  minimum volume transformation (MVT, Craig, 1994) and the archetypal analysis (AA) algorithm in Javadi and Montanari (2020). MVT is the prototype of most recent anchor-free approaches, and AA is a recent optimization approach. The results for $K=3$, $n=1000$ and 5% of labeling ratio, along with five different values of $\sigma$ (noise standard deviation), are presented in the table below. We have also included our method and SP.
>
> | Method | Ours | SP | MVT | AA |
> | --- | --- | --- | --- | --- |
> | $\sigma=0.2$ | 0.053  | 0.319 |   0.191 | 1.863 |
> | $\sigma=0.4$ |  0.064 |  1.712 | 0.380  | 4.095 |
> | $\sigma=0.6$ | 0.231  | 4.438 |  0.390 | 7.800 |
> | $\sigma=0.8$ | 0.297 | 8.404 | 0.427  |  12.521 |
> | $\sigma=1$ |  0.416 | 13.37 |  0.522 | 18.925  |
>
> We found that the optimization approaches (such as AA) were less stable; and the error rate could be really large under a high-noise level (in this case, the algorithm may be hard to converge). Our method outperforms these competitors.
>
> **5. Combination with optimization-based VH.**
>
> *"The paper mentions that the SSVH method can be combined with optimization-based VH, but this idea is left as future work. A small experiment on this combination would strengthen the practical value."*
>
> We mentioned two ways of combination: (1) Using our method as the initialization. (2) Using the $b$ estimated from our method to modify the constraints in an optimization-based method. Due to the short time, we only implemented the first idea.
>
>
> We focus on the MVT algorithm and consider two initializations: (1) Constant initialization: Set the barycentric coordinate for each point to be $w_i=(1/K){\bf 1}_K$. (2) Using our estimated vertices as the initialization. In the same simulation settings as in Point 4 above with $\sigma=0.2$, we obtain the following results:
>
> | Method | Error |
> | --- | -- |
> |Ours | 0.053 |
> MVT (constant initialization + 4 iterations) | 0.255 |
> MVT (constant initialization + 8 iterations) | 0.177 |
> MVT (our initialization + 4 iterations) | 0.114 |
> MVT (our initialization + 8 iterations) | 0.223 |
>
> When we only run 4 iterations, our initialization is better than the constant initialization. However, as the number of iterations increases to 8, the performance becomes worse. It seems that the true $V$ is not a local minimum of MVT: Even if we start from a very good initialization, the algorithm still deviates from it in just a few iterations. We should probably try a different optimization-based method other than MVT, and we will continue to explore this direction.
>
> **6. Other promising domains where SSVH could be applied.**
>
> Our method can be potentially applied to semi-supervised NMF problems. Suppose we have an NMF problem and know a few rows of one of the two factors. We can apply spectral decomposition and then apply the SCORE normalization (Jin, 2015) to eigenvectors. This will create a simplex structure, in which the barycentric coordinates and the original rows are connected as in Model (2). This is where our SSVH can be useful.
>
> **7. Unmodified real datasets.**
>
> *"The empirical section could be strengthened by including more real-world datasets ... it would be valuable to show performance on unmodified real datasets where only partial labels are available."*
>
> We absolutely agree with you, but it is hard to find such real data sets. Take the network problem for example. It is not hard to find data sets where partial labels are available (e.g., based on our prior knowledge on those high-degree nodes). However, in order to evaluate the error rates of different methods, we need to know the true labels of all nodes. Available data sets only provide categorial labels for all nodes, so we can only use them for evaluating community detection, but not mixed membership estimation. This is why we chose to do semi-synthetic experiments.

---

> > ### Comment · Reviewer_AVSi · 2025-08-05
> >
> > Thank you for your response. I will keep my score.

---

> > > ### Author Response · Authors · 2025-08-06
> > > **Reply to Reviewer AVSi**
> > >
> > > Thank you for your response. If you have any further questions or suggestions, please don’t hesitate to let us know.

---

### Official Review · Reviewer_EEzx · 2025-07-01

**Clarity:** 3
**Significance:** 2
**Originality:** 3
**Rating:** 4
**Confidence:** 2

**Summary:**

The paper addresses the "vertex hunting" problem. This is the problem of identifying the corners of a simplex, given observations consisting of points in the simplex. The particular variant studied in the paper is a semi-supervised version of the problem, where coordinates are provided for a subset of the observations, however, the observed coordinates are given in a different format. The problem formulation also allows for noise in the observations. The main contribution of the paper is the novel problem formulation and a method for solving the problem. It is also shown how to use the formulation in applications of community detection in network analysis and in topic modeling for document analysis.

**Questions:**

I would like to see further justification about the motivation of the problem, in particular about the exact formulation of the semi-supervised setting, involving the unknown vector b.

**Ethical Concerns:**

["NO or VERY MINOR ethics concerns only"]

**Final Justification:**

My main question was about the motivation of the problem, and the authors provide a satisfactory answer in the rebuttal. I upgrade my score to borderline accept, but my confidence is low, and I am not willing to champion the paper in the discussion.

**Limitations:**

There is a minimal discussion on limitations on Appendix H which is rather discussion on future work.

**Paper Formatting Concerns:**

I have no concerns about the paper formatting.

**Quality:**

3

**Strengths And Weaknesses:**

The main strengths of the paper are the following:
S1. A new interesting problem formulation is introduced for vertex hunting in a semi-supervised setting.
S2. The problem is solved using a tools from linear algebra. The error-free as well as the noisy version of the problems are analyzed and closed-form solutions are given.
S3. The proposed algorithm is analyzed theoretically, error bounds are provided, and sensible assumptions on the input data are identified for which the theoretical results hold.
S4. Interesting use cases are identified and tested empirically in network analysis and document analysis.
The main weakness of the paper is that:
W1. the problem is not so well motivated. In particular, the problem of connecting the observed coordinates \pi_i with the barycentric coordinates w_i, via the vector b, seems quite artificial. In some sense the problem seem a bit reversed engineered to fir the solution, rather than the other way around.

---

> ### Author Rebuttal · Authors · 2025-07-26
>
> Thank you for a nice summary of the four strengths of our paper. There is only one weakness you raised: how to motivate Model (2). We clarify that this model naturally arises from applications and has been known in the literature.
>
> **Where this model came from:** Both mixed-membership estimation and topic modeling essentially assume a nonnegative  factorization structure on the expectation of the data matrix. Under such structures, for any low-dimensional linear projection of data (including the spectral projection), the resulting points are contained in a simplicial cone, subject to noise corruption. To enable downstream estimation procedures for these applications,  we must first normalize this simplicial cone to a simplex (e.g., for spectral projections, the SCORE normalization (Jin, 2015) is a convenient choice). Model (2) is a direct consequence of such normalizations. For details, please see the proofs of Theorem 4.2 and Theorem 4.3, in which we verify that Model (2) indeed holds for these two applications.
>
> Model (2) was already discovered in the literature of unsupervised learning, such as Jin et al. (2024) for mixed membership estimation and Ke and Wang (2024) for topic modeling.
>
> **Why this model was not highlighted in the previous literature:** For unsupervised VH, as long as each point is contained in the simplex, we don't care how its barycentric coordinate is related to the true $\pi_i$. This is why the previous literature didn't highlight Model (2). However, for semi-supervised VH, since we need to incorporate the known $\pi_i$'s, the connection between $\pi_i$ and $w_i$ does matter and needs to be highlighted.
>
> In summary, Model (2) was not proposed by us. It was already discovered in the previous literature, although not highlighted there.

---

> > ### Comment · Reviewer_EEzx · 2025-08-03
> >
> > Thank you for the clarification to my question.

---

> > > ### Author Response · Authors · 2025-08-06
> > > **Reply to Reviewer EEzx**
> > >
> > > Thank you for your response. We’re glad to hear that our rebuttal answered your questions. If you have any further questions or suggestions, please don’t hesitate to let us know.

---

### Official Review · Reviewer_qLyi · 2025-07-03

**Clarity:** 3
**Significance:** 3
**Originality:** 3
**Rating:** 5
**Confidence:** 3

**Summary:**

The paper proposes a vertex hunting algorithm in semi-supervised setting where the membership information of a few number of data points/nodes are known. This is an extension to unsupervised vertex hunting algorithms which typically rely on stringent conditions to work. The idea stems from utilizing the projection matrices derived from the membership vectors and hence proposing an optimization-free estimation of the vertices. This also allows them to provide nice theoretical support to their approach. Synthetic and semi-synthetic experiments in community detection and topic modeling are presented to evaluate the proposed approach.

**Questions:**

Questions/Comments:

1.	Vertex hunting has close connections to structured matrix factorization (SMF) and nonnegative matrix factorization (NMF).

2.	Model 2 is the backbone of the proposed framework. But, in my opinion, a solid justification for this model is not presented. Why is Model 2 is intuitive in real settings?

3.	The uniqueness result in Theorem 2.2 looks very stringent. Under what settings $M(\alpha)$ is a one-dimensional subspace? The identifiability (uniqueness) conditions of SP algorithm, for instance, can have nice practical interpretation in community detection and topic modeling. But, here, such interpretation is missing.

4.	Also, $\Sigma(\alpha)$ needs to be $K-1$-dimensional. Under what conditions, it can be satisfied or violated in practical settings. Some discussion and intuitions might help to understand the strictness of these conditions.

5.	Theorem 4.1 and Theorem 4.2 demand an exact recovery. Do these theorems require the uniqueness conditions in Theorem 2.2 to be satisfied?

6.	In the experiments, do the algorithms SP and SVS utilize the labeled information using some sort of initialization? It may also be worthwhile to initialize the SP/SVS algorithms or other NMF algorithms using volume minimization to initialize with your approach and observe the empirical performance.

7.	Some minor comments:

a.	V is not introduced in line 68

b.	$\Pi_S$ is not explicitly defined in Eq. (3)

c.	in line 147, bing -> “being”

**Ethical Concerns:**

["NO or VERY MINOR ethics concerns only"]

**Final Justification:**

The authors have addressed my concerns over some of the theoretical claims and questions over initializing the algorithm in practically feasible way. Overall, I liked the idea and how the work promotes the structured matrix factorization for weekly supervised settings. Hence, I keep my score on accepting the paper.

**Limitations:**

yes

**Quality:**

4

**Strengths And Weaknesses:**

Strengths:

1.	The paper is well-written with nice theoretical support of the proposed framework.

2.	The work can inspire follow up research in the field of structured matrix factorization and nonnegative matrix factorization in semi-supervised, noisy settings.

Weaknesses:

1.	Limited empirical evaluation and comparison. There are many recent community detection and topic modeling methods. Only SP and SVS are employed in the study.

---

> ### Author Rebuttal · Authors · 2025-07-31
>
> Thank you for your insightful comments and positive evaluation! We are especially glad that you think our work can "inspire follow-up research in the field of structured matrix factorization and nonnegative matrix factorization." Please find below our point-to-point response.
>
> **1. Empirical evaluation and comparison.**
>
> *"There are many recent community detection and topic modeling methods. Only SP and SVS are employed in the study."*
>
> In these applications, VH is used as one plug-in step. Although there are different methods, many of them actually use SP in their VH step. For this reason, we hope to find methods that don't use SP in the VH step. Inspired by the comment of  another reviewer, we found that Huang et al. (2016) "Anchor-free correlated topic modeling: Identifiability and algorithm" gave such an algorithm. We implemented it on the semi-synthetic data with calibrated paraders from the paper abstracts in MADStat, where we added $12$ labeled words. The $L_1$-estimation error on the topic matrix is $0.023$ for our method and $0.030$ for the method in Huang et al. (2016).
>
> Additionally, we compared with other NMF-based unsupervised methods in the sub-Gaussian-noise setting. This includes the minimum volume transformation (MVT, Craig, 1994) and the archetypal analysis (AA) algorithm in Javadi and Montanari (2020). The results for $K=3$, $n=1000$ and 5% of labeling ratio, along with five different values of $\sigma$ (noise standard deviation), are presented in the table below. We have also included our method and SP.
>
> | Method | Ours | SP | MVT | AA |
> | --- | --- | --- | --- | --- |
> | $\sigma=0.2$ | 0.053  | 0.319 |   0.191 | 1.863 |
> | $\sigma=0.4$ |  0.064 |  1.712 | 0.380  | 4.095 |
> | $\sigma=0.6$ | 0.231  | 4.438 |  0.390 | 7.800 |
> | $\sigma=0.8$ | 0.297 | 8.404 | 0.427  |  12.521 |
> | $\sigma=1$ |  0.416 | 13.37 |  0.522 | 18.925  |
>
> We found that the optimization approaches (such as AA) were less stable; and the error rate could be really large under a high-noise level (in this case, the algorithm may be hard to converge). Our method outperforms these competitors.
>
> **2. Justification of Model (2).**
>
> For the network application, we justify Model (2) in Theorem 4.1. In detail, let $U$ be an arbitrary $n\times K$ projection matrix, and let $\eta\in\mathbb{R}^K$ be an arbitrary vector. We construct the following $K$-dimensional points:
> $$\hat{r}_i = U'Ae_i/\eta'U'Ae_i, \qquad 1\leq i\leq n.$$
>
> Here, $Ae_i$ is the $i$th column of $A$, and $U'Ae_i$ is a projection of this column into low dimensionality, and the denominator $\eta'U'Ae_i$ serves as a normalization. In a special case where $U$ consists of the first $K$ eigenvectors of $A$ and  $\eta=(1,0, \ldots, 0)'$, the points $\hat{r}_i$ are called the SCORE projections (Jin, 2015) in the literature, and these points are used for the downstream estimation procedure. In Theorem 4.1, we consider a population counterpart of $\hat{r}_i$ as follows: Let $\Omega=\mathbb{E}A$ be the Bernoulli probability matrix. Consider
> $$r_i = U'\Omega e_i/\eta'U'\Omega e_i, \qquad 1\leq i\leq n.$$
>
> In Theorem 4.1 Bullet Point (a), we show that if $\Omega$ satisfies the degree-corrected mixed membership (DCMM) model, then there exists a matrix $V$ and a vector $b$ such that
> $$r_i = Vw_i, \quad \mbox{with}\quad w_i = (b\circ \pi_i)/\|b\circ \pi_i\|_1.$$
>
> This is exactly Model (2).
>
> For the topic modeling problem, we have a similar justification in the proof of Theorem 4.2 (due to the very tight space constraint, we didn't state this explicitly in Theorem 4.2, but it was indeed in the proof).
>
> We also clarify that Model (2) is not new. It has been discovered in the previous literature. However, since the connection between $\pi_i$ and $w_i$ does not matter for unsupervised VH, this model was not highlighted in the previous literature.
>
> **3. The uniqueness result in Theorem 2.2 and the conditions on $\Sigma(\alpha)$.**
>
> *"Under what settings $M(\alpha)$ is a one-dimensional subspace?"*
>
> By our Theorem 2.2, $M(\alpha)$ has a one-dimensional null space if $\Sigma(\alpha)$ has a rank $(K-1)$.
>
> *"$\Sigma(\alpha)$ needs to be $(K-1)$-dimensional. Under what conditions, it can be satisfied or violated in practical settings."*
>
> In Line 123, we mentioned that $\Sigma(\alpha)$ is a weighted sample covariance matrix of $w_i$'s for $i\in S$. Since $w_i$'s live in a $(K-1)$-dimensional hyperplane, it is reasonable to assume that the sample covariance matrix has a rank $(K-1)$. For illustration, let's tentatively assume that $b={\bf 1}_K$ and $H\alpha={\bf 1}_n$. Below are two examples for which $\Sigma(\alpha)$ has a rank $(K-1)$:
> - $S$ only contains $(K+1)$ points, with one point at each vertex and the last point located at the center of the simplex (defined as $K^{-1}\sum_{k=1}^K v_k$). We can theoretically show that  $\Sigma(\alpha)$ has a rank $(K-1)$.
> - For the labeled points, their $\pi_i$'s are i.i.d. sampled from a Dirichlet distribution, with the Dirichlet parameters being all positive constants (in this case,  as $N\to\infty$, $\Sigma(\alpha)$ has a rank $(K-1)$ with an overwhelming probability).
>
> **4. Using our method to initialize other algorithms**
>
> *"In the experiments, do the algorithms SP and SVS utilize the labeled information using some sort of initialization? It may also be worthwhile to initialize the SP/SVS algorithms or other NMF algorithms using volume minimization to initialize with your approach and observe the empirical performance."*
>
> SP and SVS do not need initializations. Using our method to initialize MVT is an interesting idea. With the same simulation settings as in Point 1 above with $\sigma=0.2$, we compared two ways of initializing MVT: (1) Constant initialization: Set the barycentric coordinate for each point to be $w_i=(1/K){\bf 1}_K$. (2) Using our estimated vertices as the initialization.
>
> | Method | Error |
> | --- | -- |
> |Ours | 0.053 |
> MVT (constant initialization + 4 iterations) | 0.255 |
> MVT (constant initialization + 8 iterations) | 0.177 |
> MVT (our initialization + 4 iterations) | 0.114 |
> MVT (our initialization + 8 iterations) | 0.223 |
>
> When we only run 4 iterations, our initialization is better than the constant initialization. However, as the number of iterations increases to 8, the performance becomes worse. It seems to suggest that the true $V$ is not a local minimum of MVT: Even if we start from a very good initialization, the algorithm still deviates from it in just a few iterations.

---

> > ### Comment · Reviewer_qLyi · 2025-08-07
> >
> > Thank you for your response and addressing my comments. I choose to keep the score.

---

### Official Review · Reviewer_GfMr · 2025-07-08

**Clarity:** 2
**Significance:** 2
**Originality:** 3
**Rating:** 4
**Confidence:** 4

**Summary:**

This paper introduces semi-supervised vertex hunting (SSVH), an extension of the classical vertex hunting problem that incorporates partial label information. The key technical contribution is establishing an interesting connection between the eigenvector of zero eigenvalue for a constructed matrix and the unknown parameter $b$, leading to a clean optimization-free approach for estimating $b$. The paper provides theoretical guarantees showing improved convergence rates compared to unsupervised methods and demonstrates applications in network mixed membership estimation and topic modeling. The method is validated on both synthetic and real datasets, with empirical results showing consistent improvement over baseline methods.

**Questions:**

1. For Figure 2, why does SP perform equally poorly even in the case with pure points? Also, error bars should be added since results are based on 100 repetitions.

2. In line 344, the paper claims that 5% labels is a "small proportion". However, in many semi-supervised learning contexts, 5% is not typically considered small. Can the authors justify why they consider this a small proportion?

3. The paper should compare with state-of-the-art anchor-free approaches that don't require pure nodes. While the NeurIPS 2016 paper is one example, there are more recent developments in this area that should be considered.

4. The sub-gaussian noise assumption needs justification in the context of specific applications. What is the nature of noise in network mixed membership estimation and topic modeling? Is this assumption reasonable in these settings?

5. In real-world applications, there is likely to be noise in the labels or known membership vectors themselves. How does label noise affect the theoretical guarantees?

**Ethical Concerns:**

["NO or VERY MINOR ethics concerns only"]

**Final Justification:**

After careful consideration of the paper, reviews, and author rebuttal, I revise my recommendation to borderline accept.

Issues Resolved:
1. Theoretical framework and proofs were clarified and strengthened with noise assumptions
2. Additional baselines (MVT, AA) were added for comparison

Issues Remaining Unresolved:
1. While a faster rate than unsupervised methods is shown, there's no proof of optimality or lower bound for semi-supervised setting, making it hard to justify the claimed difficulty of achieving such rates
2. Despite adding comparisons with anchor-free methods, the possibility of adapting these methods to leverage labeled points is not thoroughly addressed
3. Limited real-world validation as experiments rely heavily on synthetic/semi-synthetic data

**Limitations:**

Yes

**Quality:**

2

**Strengths And Weaknesses:**

### Strengths:

- The paper is technically sound with complete proofs and clear assumptions.
- The connection between the eigenvector of zero eigenvalue for the constructed matrix $M(\alpha)$ and the unknown transformation vector $b$ is interesting, leading to a clean optimization-free approach.
- The paper provides good reproducibility with code provided.

### Weaknesses:

- For vertex hunting problems, having information about the position/membership of even a few points naturally makes the identification problem much easier, so the improved performance is not surprising.
- The theoretical analysis focuses on the entire $V$ matrix, while recent literature emphasizes row-wise bounds.
- The literature review is insufficient and misses critical prior work. For example, "Anchor-free Correlated Topic Modeling: Identifiability and Algorithm" in NeurIPS 2016 and its follow-ups are not discussed, despite being highly relevant as they solve similar problems without requiring pure nodes.
- The paper lacks theoretical guarantees specific to the applications (network mixed membership and topic modeling)
- The notations are messy and sometimes confusing, making it difficult to follow the transitions between different problem settings.
- The empirical evaluation could be more comprehensive with additional modern baselines, and some experimental results need to be clarified or simplified for better interpretation (will raise in Questions session).

---

> ### Author Rebuttal · Authors · 2025-07-25
>
> Thank you for your comments. We are glad that you think our discovery of the optimization-free approach is interesting and that our paper is technically solid.
>
> **1. Comparison with the literature, especially the anchor-free approach.**
>
> Since several of your critical comments are related to this, we hope to clarify this point first. Thanks for pointing out the anchor-free paper, and we will add a nice citation. However, our work is significantly different from it:
> - *Different settings*: The anchor-free approach is for the unsupervised setting, but our focus is the semi-supervised setting. The anchor-free approach is not able to leverage the partial label information.
> - *Different methods*: The anchor-free approach solves a non-convex optimization and requires an iterative algorithm to solve, while our approach is optimization-free.
> - *Different error rates*: All unsupervised methods, including the anchor-free approach, can only achieve an error rate of $\widetilde{O}(1)$. However, our semi-supervised method is able to achieve a faster rate of $\widetilde{O}(N^{-1/2})$, where $N$ is the number of labeled points.
>
> In summary, although anchor-free is one advantage of our method, it is not our main goal. Our main goal is to leverage the labeled points to achieve a "faster" error rate than all unsupervised methods. Our contribution is orthogonal to those of the anchor-free approaches.
>
> **2. Additional baselines.**
>
> We added two NMF-based unsupervised methods: the minimum volume transformation (MVT, Craig, 1994) and the archetypal analysis (AA) algorithm in Javadi and Montanari (2020). They are both anchor-free approaches: MVT bypasses the anchor-point condition by minimizing the volume of the simple, and AA bypasses it by minimizing the distance from vertices to the convex hull of data points.
>
> The results for $K=3$, $n=1000$ and 5% of labeling ratio, along with five different values of $\sigma$ (noise standard deviation), are presented in the table below. We have also included our method and SP.
>
> | Method | Ours | SP | MVT | AA |
> | --- | --- | --- | --- | --- |
> | $\sigma=0.2$ | 0.053  | 0.319 |   0.191 | 1.863 |
> | $\sigma=0.4$ |  0.064 |  1.712 | 0.380  | 4.095 |
> | $\sigma=0.6$ | 0.231  | 4.438 |  0.390 | 7.800 |
> | $\sigma=0.8$ | 0.297 | 8.404 | 0.427  |  12.521 |
> | $\sigma=1$ |  0.416 | 13.37 |  0.522 | 18.925  |
>
> We found that the optimization approaches (such as AA) were less stable; and the error rate could be really large under a high-noise level (in this case, the algorithm may be hard to converge). Our method outperforms these competitors.
>
> **3. Numerical comparison with the 2016 NeurIPS paper.**
>
> Thanks for pointing out this interesting reference. Before presenting the numerical results, we'd like to clarify two points:
>  - *Their method is not for a general VH setting*: In their Equation (7), the constraints  are that each column of the topic matrix is nonnegative and self-normalized. These constraints do not hold for a general VH problem.
> - *An adaptation of their method to a general VH setting reduces to Minimum Volume Transformation (MVT)*: In a general VH setting, we may adapt their method by keeping the same objective but removing those problem-specific constraints. Their objective is maximizing the determinant of $M=V^{-1}$, which is equivalent to minimizing the volume of the simplex spanned by $V$. This becomes the classical MVT algorithm (which has already been discussed in our Remark 1 and Figure 1).
>
> Since this paper didn't propose a general VH algorithm, we can only compare it in the topic model setting. We use the MADStat data set to calibrate a true topic model and then simulate data from it. We pick only $N=12$ labeled points (words). The mean and standard deviation of the L1-error of topic modeling over 100 repetitions are reported below.
>
> | Method | Ours  | Anchor-free |
> | ------ | ------ | ----- |
> | **Error**  | 0.023 (0.005)  | 0.030 (0.001) |
>
> **4. Labeling ratio.**
>
> *"In line 344, the paper claims that 5% labels is a 'small proportion.' However, in many semi-supervised learning contexts, 5% is not typically considered small. Can the authors justify why they consider this a small proportion?"*
>
> We will edit the text not to claim 5% as a 'small proportion.' In theory, our method can work with as few as $(K+1)$ labeled points. Our numerical results also contain low-labeling-ratio cases:
> - In the left two plots of of Figure~2, the x-axis is the labeling ratio, ranging from 1% to 5% on a fine grid. Our method outperforms SP in  this whole range. Since $n=1000$ in these experiments, a 1% labeling ratio means there are only $N=10$ labeled points.
> - In our newly added topic modeling experiment (see Point 3 above), we only pick $N=12(=K+1)$ labeled points.
>
> **5. Justification of the sub-Gaussian noise assumption in applications.**
>
> This is a great point. We recall that in these applications, VH algorithms are applied to leading eigenvectors of data matrices. Recent results have shown that the entries of leading eigenvectors are approximately normal. To see the rationale, we take the network setting for example. We quote the result in Abbe et al. (2020) "Entrywise eigenvector analysis of random matrices with low expected rank". Let $A$ be the n-by-n network adjacency matrix, and let $\hat{\xi}_k$ and $\xi_k$ be the respective $k$th eigenvector of $A$ and $\mathbb{E}A$.  Under the stochastic block model (SBM), Abbe et al. (2020) showed that $$\hat{\xi}_k= A\xi_k + \delta_k,$$
>
> where $\delta_k$ is an entry-wise negligible vector, and $A\xi_k$ is called the first-order proxy of $\hat{\xi}_k$. Each entry of $A\xi_k$ is a weighted linear combination of $n$ entries of $A$ (these are independent Bernoulli variables), hence being sub-Gaussian. Since the entries of $\delta_k$ are at a smaller order,  each entry of $\hat{\xi}_k$ is also approximately sub-Gaussian.
>
> **6. How does label noise affect the theoretical guarantees?**
>
> Yes, our theory can be modified for this setting. Let $\Pi_S$ and $\widehat{\Pi}_S$ be the true and noisy label matrices for those labeled points. There will be an extra term in the error bound (the norm below is the spectral norm):
> $$N^{-1/2}\| \widehat{\Pi}_S-\Pi_S \|.$$
>
> Notably, this term will not explode as $N$ increases, so our method still enjoys a good error rate.
>
> **7. Row-wise bounds.**
>
>  *"The theoretical analysis focuses on the entire $V$ matrix, while recent literature emphasizes row-wise bounds."*
>
> Our results are in fact stronger than row-wise bounds. Recall that $V=[v_1, v_2,\ldots,v_K]$. Let $e_k$ denote the $k$ th standard basis of $\mathbb{R}^K$. It is seen that for each $1\leq k\leq K$,
>
> $$\| \hat{v}_k-v_k \| = \|(\widehat{V}-V) e_k\| \leq  \text{SpectralNorm}(\widehat{V}-V).$$
>
> Since our theory bounds the spectral norm of $\widehat{V}-V$, it automatically implies the row-wise bounds.
>
>
> **8. Why SP has a worse performance even with pure points.**
>
> *"For Figure 2, why does SP perform equally poorly even in the case with pure points? Also, error bars should be added since results are based on 100 repetitions."*
>
> As we mentioned in Point 1 above, anchor-free is only one of the advantages of our method. Another advantage is the improved error rate by a factor of $1/\sqrt{N}$. In the pure-point case of Figure 2, the x-axis is the labeling ratio. Even with 1% (corresponding to $N=10$) of labeled points, the theoretical error rate is only $1/\sqrt{10}\approx 0.32$ times the theoretical error rate of SP. This explains the results in Figure 2. The error rate is indeed a fundamental difference between unsupervised and semi-supervise settings.
>
> Yes, we will add error bars in the revision. Thanks for this suggestion.
>
> **9. Other clarifications.**
>
> *"Having information ... makes the identification problem much easier, so the improved performance is not surprising."*
>
> Although people expect the semi-supervised setting to be easier than the unsupervised setting, how to design an algorithm that indeed achieves a faster rate is not easy. Both our method and theory are non-trivial. For example, we explicitly show that leveraging the labeled points will strictly improve the error rate, which has never been discovered before.
>
> *"The paper lacks theoretical guarantees specific to the applications"*
>
> We did provide theorems for both applications. Please see Theorem 4.1 and Theorem 4.2.

---

> > ### Comment · Reviewer_GfMr · 2025-08-09
> >
> > After careful consideration of the paper, reviews, and author rebuttal, I revise my recommendation to borderline accept.
> >
> > Issues Resolved:
> >
> > - Theoretical framework and proofs were clarified and strengthened with noise assumptions
> > - Additional baselines (MVT, AA) were added for comparison
> >
> > Issues Remaining Unresolved:
> >
> > - While a faster rate than unsupervised methods is shown, there's no proof of optimality or lower bound for semi-supervised setting, making it hard to justify the claimed difficulty of achieving such rates
> > - Despite adding comparisons with anchor-free methods, the possibility of adapting these methods to leverage labeled points is not thoroughly addressed
> > - Limited real-world validation as experiments rely heavily on synthetic/semi-synthetic data

---

> ### Author Response · Authors · 2025-08-09
> **Reply to Reviewer GfMr**
>
> Thank you for your response. We are glad to hear that our rebuttal addressed some of the issues you raised, and we sincerely appreciate your decision to raise the score.
>
> Below please find our quick answers to your "Issues Remaining Unresolved":
>
> **Optimality of the Rate:** In the paper, we considered an Ideal Estimator which assumes that $b$ is known. In this case, we can solve the vertex matrix by a simple least-squares, and its rate is presented in Lemma 3.3. We showed that our rate matches with this rate of the Ideal Estimator up to a $\sqrt{\log(n)}$ factor. While this is not a lower bound, it provides some evidence that our rate is hard to improve (because when noise is Gaussian, least-squares is hard to improve).
>
> **Adapting anchor-free methods to semi-supervised settings:** As discussed in the paper, we are not aware of any semi-supervised variant of these methods. Instead, our work provides such a solution: We can either (1) use our $\widehat{V}$ to initialize another method, or (2) use our $\hat{b}$ to modify the constraints/objectives of another method. In both approaches, information of labeled nodes is incorporated.
>
> During the rebuttal period, we were only able to experiment on Approach (1). The results were included in our rebuttal to other reviewers, in which we used our estimator to initialize MVT.
>
> After a second thought, we realize that Approach (2) should be the more promising one, as a modified constraint/objective will impact *all iterations*, not just the initialization. In detail, let $\hat{w}_i = (\hat{b}\circ \pi_i)/\|\hat{b}\circ \pi_i\|_1$ for all labeled nodes $i$, where $\hat{b}$ is our estimator of $b$. We hope to enforce the constraint that $x_i\approx V\hat{w}_i$. This can be done by adding a penalty in the objective: Let $L(V)$ be the original loss. The new loss is defined by
>
> $$L(V) + \lambda \sum_{i \in S} (x_i - V\hat{w}_i)^2.$$
>
> We'd be glad to explore this idea. However, a thorough study of this will demand significant empirical/theoretical efforts, and we prefer to defer it to future work.

---

### Author Response · Authors · 2025-08-08
**Comment to All Reviewers**

Dear Reviewers,

Thank you for your valuable feedback on our paper. As the author–reviewer discussion period is about to close, we would like to check whether you have any additional questions or comments. We will be happy to respond promptly.

Based on the feedback received so far, we plan to make the following revisions:
- **Literature Review:** We will incorporate the references suggested by the reviewers. In particular, we will expand the literature review on unsupervised vertex hunting algorithms, including both anchor-free methods and approaches from the structured NMF literature.
- **Motivation of Model (2):** We will clarify that this model has appeared in the literature (with detailed references) and note that it is also proved in the supplement for the two applications we study.
- **Numerical Experiments:** We will include the new experiments conducted during the rebuttal period, such as comparisons with additional unsupervised NMF algorithms and anchor-free approaches, robustness analyses with respect to the choice of $\alpha$, runtime analysis, and a hybrid approach combining our method with other optimization-based techniques.
- **Clarifications and Remarks:** We will add remarks and clarifications to address reviewer comments (e.g., extensions to noisy/imperfect labels, interpretation of the conditions in Theorem 2.2, and how our current results imply row-wise bounds).

Since the above points were already studied in our rebuttal, incorporating them into the revision will be relatively straightforward.

Finally, we wish to highlight that our paper is the first to rigorously study the semi-supervised setting for the Vertex Hunting problem. We do not view the unsupervised VH literature as competing work; rather, the two lines of research have distinct focuses and complementary strengths.

We hope our planned revisions will address your concerns and that you find our responses satisfactory. Thank you again for your insightful feedback and constructive suggestions.

Sincerely,

Authors of Submission 13861

---

### Note · Authors · 2025-08-11

Dear Area Chair and Reviewers,

We sincerely thank all of you for the thoughtful comments and inspiring discussions. We'd like to take this opportunity to add two remarks.

-------

### 1. A new insight.

We highlight a new insight prompted by the reviewers' comments: Our method actually "enables" the extension of many unsupervised Vertex Hunting (VH) approaches to semi-supervised settings. This is achieved by the following two-step procedure:
- Step 1 (**core idea**): Apply our optimization-free approach to estimate $b$, and use $\hat{b}$ to compute the barycentric coordinate $\hat{w}_i$ for each labeled node.
- Step 2 (**flexible**): Minimize $L(V) +\lambda \sum_{i\in S} \|x_i- V\hat{w}_i\|^2$, where $L(V)$ is the loss function of an unsupervised VH method.

The current method is a special case with $\lambda=\infty$, in which Step 2 reduces to the simple least squares. From this perspective, our proposed approach is more general and broadly useful than we initially realized.

PS: This discovery emerged after several reviewers encouraged us to combine our approach with unsupervised methods. Our initial attempt—using our method as an initialization for MVT—yielded only marginal improvement. We then realized that instead of serving merely as an initialization, our method should directly modify the optimization objective, ensuring that labeled-node information is used in every iteration. This insight led us to the general framework described above.

-------

### 2. Semi-synthetic experiments.

We briefly explain why we did semi-synthetic experiments. In mixed-membership estimation, unlike in community detection, it is difficult to find datasets with ground-truth labels for all nodes (rather than just a small subset of nodes). While we could present results on real data, we would not be able to report estimation errors. Reporting such errors is important for demonstrating the benefit of leveraging labeled nodes.

We acknowledge that relying on semi-synthetic data is a limitation; however, in this context, it is a practical choice, as real data can still be used to calibrate model parameters, and the subsequent semi-synthetic experiments enable us to clearly quantify the gains.

-------

We greatly appreciate the reviewers’ comments and suggestions, which have inspired several new insights and helped us improve the presentation of our work!

Sincerely,

Authors of Submission 13861

---

### Decision · Program_Chairs · 2025-09-17

**Decision:**

Accept (poster)

**Comment:**

The paper introduces semi-supervised vertex hunting (SSVH), where some data points come with barycentric coordinates known only up to an unknown transformation. It proposes a linear-algebraic method with provable guarantees built on properties of orthogonal projection matrices to estimate the underlying simplex. The authors present practical, scalable algorithms for mixed-membership networks and topic modeling. The reviewers appreciated the solid theoretical guarantees and the clear exposition. For the camera-ready, the authors should incorporate the rebuttal clarifications and added experiments, and expand the unsupervised VH literature review. I am including some references from my discussion with some of the reviewers here -
1. “Detecting overlapping communities in networks using spectral method”, SIAM Journal of Mathematics of Data Science, 2020 (see citations 3, 30, 31, 33, 39–41 in the journal version);
2. “Anchor-Free Correlated Topic Modeling: Identifiability and Algorithm,” NeurIPS 2016;
3. “A statistical interpretation of spectral embedding: the generalized random dot product graph,” JRSS-B, 2022.